

# Explaining detection heterogeneity with finite mixture and non-Euclidean movement in spatially explicit capture-recapture models

Robby R. Marrotte[1], Eric J. Howe[1], Kaela B. Beauclerc[1], Derek Potter[1] and Joseph M. Northrup[1,2]

[1] Wildlife Research & Monitoring Section, Ministry of Northern Development, Mines, Natural Resources and Forestry, Peterborough, Ontario, Canada
[2] Environmental and Life Sciences Graduate Program, Trent University, Peterborough, Ontario, Canada

Corresponding author
Robby R. Marrotte,
robbymarrotte@trentu.ca

## ABSTRACT

Landscape structure affects animal movement. Differences between landscapes may induce heterogeneity in home range size and movement rates among individuals within a population. These types of heterogeneity can cause bias when estimating population size or density and are seldom considered during analyses. Individual heterogeneity, attributable to unknown or unobserved covariates, is often modelled using latent mixture distributions, but these are demanding of data, and abundance estimates are sensitive to the parameters of the mixture distribution. A recent extension of spatially explicit capture-recapture models allows landscape structure to be modelled explicitly by incorporating landscape connectivity using non-Euclidean least-cost paths, improving inference, especially in highly structured (riparian & mountainous) landscapes. Our objective was to investigate whether these novel models could improve inference about black bear (*Ursus americanus*) density. We fit spatially explicit capture-recapture models with standard and complex structures to black bear data from 51 separate study areas. We found that non-Euclidean models were supported in over half of our study areas. Associated density estimates were higher and less precise than those from simple models and only slightly more precise than those from finite mixture models. Estimates were sensitive to the scale (pixel resolution) at which least-cost paths were calculated, but there was no consistent pattern across covariates or resolutions. Our results indicate that negative bias associated with ignoring heterogeneity is potentially severe. However, the most popular method for dealing with this heterogeneity (finite mixtures) yielded potentially unreliable point estimates of abundance that may not be comparable across surveys, even in data sets with 136–350 total detections, 3–5 detections per individual, 97–283 recaptures, and 80–254 spatial recaptures. In these same study areas with high sample sizes, we expected that landscape features would not severely constrain animal movements and modelling non-Euclidian distance would not consistently improve inference. Our results suggest caution in applying non-Euclidean SCR models when there is no clear landscape covariate that is known to strongly influence the movement of the focal species, and in applying finite mixture models except when abundant data are available.

## INTRODUCTION

Monitoring wildlife populations is fundamental to conserving threatened or endangered species and managing game animals (*Goldsmith, 1991*). Monitoring allows practitioners to assess population trends and detect declines (*Yoccoz, Nichols & Boulinier, 2001*), determine the effectiveness of conservation or management actions (*Campbell et al., 2002*), and ensure sustainable harvest (*White et al., 2015*; *Fryxell, Sinclair & Caughley, 2014*; *Bender, 2006*; *Amstrup et al., 2005*). Logistical, financial, and practical constraints dictate that most population monitoring programs cannot count all individuals in a population since detection is imperfect and probabilistic. Imperfect detection will not bias population size estimates if detection rates for monitoring protocols are constant across time, space, and individuals, but this is rarely the case (*Howe, Obbard & Kyle, 2013*; *Sollmann et al., 2013*). Detectability of individuals may vary with age, sex, body mass, temperament, different learning experiences, including prior exposure to humans, or behaviors related to life history (*Gimenez, Cam & Gaillard, 2018*; *Guillera-Arroita, 2017*). Consequently, different individuals of the same species, and even of the same age and sex, may have intrinsically different probabilities of being detected during surveys, and the sources of heterogeneity may not be observed. Individual heterogeneity in detectability can cause bias to estimates of any metric relying on the detectability of individuals, including occupancy and abundance (*Gimenez, Cam & Gaillard, 2018*; *Guillera-Arroita, 2017*; *Kellner & Swihart, 2014*; *Efford & Dawson, 2012*; *Pledger, 2000*). Further, the existence of heterogeneity calls into question inference on second-order estimates relating to, for example, habitat use, movement, and population trends. Consequently, detection heterogeneity that is unaccounted for can cause bias and is a general problem for monitoring programs (*Kellner & Swihart, 2014*; *Kéry & Schmidt, 2008*).

Capture-recapture (CR) methods are widely used for estimating animal abundance. Several variations have been proposed to avoid or account for individual heterogeneity in detection probabilities, which causes negative bias in estimates of abundance when not modelled (*Pledger, Pollock & Norris, 2010*; *Pledger, Pollock & Norris, 2003*; *Pledger, 2000*; *Chao, 1987*; *Otis et al., 1978*). Proper study design and the inclusion of covariates of detectability (such as time of day, weather, age, and sex) can minimize unmodelled heterogeneity in CR data and, therefore, negative bias in estimates (*Gimenez, Cam & Gaillard, 2018*; *Guillera-Arroita, 2017*; *Miller et al., 2015*). However, important covariates may not be observed. For example, age remains unknown in genetic surveys, and different learning experiences of long-lived, intelligent species affect behaviour but can rarely be quantified since all sources of heterogeneity can rarely be modelled using covariates (*Thandrayen & Wang, 2009*; *Noyce, Garshelis & Coy, 2001*). Statistical methods for accounting for heterogeneity due to unobserved or unobservable covariates have been developed (*Gimenez, Cam & Gaillard, 2018*; *Gimenez & Choquet, 2010*; *Royle, 2008*; *Royle,*

*2006*; *Pledger, 2000*; *Chao, 1987*; *Otis et al., 1978*). Finite mixture models have become the most common approach to modelling heterogeneity from unobserved sources (*Pledger, 2000*). They assume that the population has two or more latent groups with different detection parameters; the proportions of animals that belong to the different groups must also be estimated (*Pledger, 2000*). However, their reliable application remains problematic (*Dorazio, 2014*; *Pledger & Phillpot, 2008*; *Link, 2004*; *Link, 2003*).

Spatial capture-recapture (SCR) models explicitly account for one important source of individual heterogeneity in capture probabilities; the placement of detectors relative to the areas where individuals spend most of their time (*i.e.*, their activity centers; *Borchers & Efford, 2008*; *Boulanger, Stenhouse & Munro, 2004*) and are now preferred over non-spatial CR models in many situations (*Greenspan, Anile & Nielsen, 2020*; *Arandjelovic & Vigilant, 2018*; *Clutton-Brock & Sheldon, 2010*; *Obbard, Howe & Kyle, 2010*). SCR models also allow other sources of heterogeneity to be modelled using covariates or mixture distributions (*Royle et al., 2009*; *Borchers & Efford, 2008*). As in CR models, it is not always clear whether a dataset has sufficient information to accurately estimate the parameters of the mixture distribution, potentially leading to more biased results than if heterogeneity was ignored (*Pledger & Phillpot, 2008*; *Link, 2004*; *Link, 2003*). In the face of such uncertainty, it is generally left to the practitioner to decide whether or how to account for unobserved sources of detection heterogeneity within an appropriate statistical framework (*Gimenez, Cam & Gaillard, 2018*; *Miller et al., 2015*; *Royle et al., 2009*). Therefore, it is preferable to explicitly model specific, observable sources of heterogeneity rather than relying on statistical approaches used to model heterogeneity due to unobserved sources. For example, when heterogeneity is modelled using finite mixture distributions, the groupings of animals into (usually only two) latent classes may have little biological meaning.

SCR models assume a monotonic decline in detection probability with distance from the activity center; the form of this relationship is often assumed to follow a half-normal probability or hazard distribution (Euclidean distance SCR models, *Royle et al., 2009*; *Borchers & Efford, 2008*). However, in heterogeneous landscapes, Euclidean distance is not a realistic metric to describe the time, cost or effort required to travel between activity centers and detectors. Euclidean distance SCR models assume that there are no features that impede or promote movement and, therefore, movement capability is similar for all individuals. However, the movement capability of nonvolant animals through structured landscapes is not uniform. Landscape features such as lakes, mountains, or human development can impede movement (landscape resistance), whereas other features offer little resistance and may be used as corridors. Many animals have a detailed spatial memory of the landscape they occupy; this allows them to reduce their movement cost and increase their fitness (*Halsey, 2016*; *Shepard et al., 2013*). As a result, animals' paths between locations are not straight, and Euclidean distance is a poor measure of the movement cost to animals.

*Royle et al. (2013)* demonstrated that assuming a monotonic decline in detectability with Euclidean distance could cause negative bias in SCR estimates of abundance in structured landscapes (where mobility of animals is affected by landscape features) and attributed the bias to unmodelled individual heterogeneity induced by the different landscapes individuals were exposed to within their home ranges. They described methods

for estimating population density by modelling detectability as a function of the movement capability of a species within the SCR framework. More specifically, they modelled the decline in detectability as a function of non-Euclidean distances. Most animals exhibit variation in movements across a landscape; thus, the ability to incorporate movement costs into SCR models is a significant development. Non-Euclidean distances are generally calculated using the Dijkstra least-cost path algorithm (*Dijkstra, 1959*). This algorithm uses a spatial layer or cost surface that assigns a cost of travelling through a pixel and finds the least-cost path between locations. SCR models that account for the non-Euclidean nature of movement cost offer an interpretable means of understanding heterogeneity rather than attributing it to unobserved sources.

*Sutherland, Fuller & Royle (2015)* previously tested the non-Euclidean SCR approach on simulated encounter histories of European otters (*Lutra lutra*) in a riparian habitat system, showing improved density estimation. Since then, two studies of black bears (*Ursus americanus*; *Morin et al., 2017*; *Murphy et al., 2016*), two studies of American mink (*Neogale vison*; *Sutherland, Fuller & Royle, 2015*; *Fuller et al., 2016*), one of jaguars (*Panthera onca*; *Tobler et al., 2018*), and one of snow leopards (*Panthera uncia*; *Sharma et al., 2020*) have used this approach to account for individual heterogeneity in detectability. In all 6 cases, the authors found support for non-Euclidean models. In most of these examples, the landscapes were highly structured, and the movements of animals were accordingly strongly influenced. Minks concentrate their movements near waterbodies (*Fuller et al., 2016*). Snow leopard habitat is characterized by extreme variation in elevation such that two-dimensional Euclidean distance is a poor measure of distance travelled (*Sharma et al., 2020*). Black bear movements in fragmented landscapes in southern New York state were facilitated by forest cover and hindered by developed areas (*Morin et al., 2017*). Black bear movements in Kentucky followed the orientation of mountain ridges (*Murphy et al., 2016*). Jaguars used roads as important movement corridors, and ignoring this effect would have led to underestimating densities (*Tobler et al., 2018*). Thus, the application of non-Euclidean models to species with dramatic or predictable effects of landscape structure on their movement is both recommended and feasible. However, additional testing of these models to identify situations in which they are most effective is still warranted because animals do not move randomly, even in less structured landscapes.

We were interested in whether non-Euclidean models could improve inference where obvious barriers and corridors were absent and where different landscape characteristics could affect the cost of movement. In this study, we applied the non-Euclidean spatial-capture recapture approach to data collected from 51 independent study areas sampled in 2017 or 2018 as part of a broad-scale black bear monitoring program in Ontario, Canada. Genetic SCR surveys have been used to monitor black bear populations in Ontario since the early 2000s; estimates inform decision-making, including harvest management (*OMNRF, 2019*). We expected the detectability of individual bears to vary, including within sexes, for the reasons described above including landscape structure, and because individual heterogeneity was apparent in SCR data from prior surveys of the female fraction of the same population (*Howe, Obbard & Kyle, 2013*; *Obbard, Howe & Kyle, 2010*); data were pooled across study areas for analysis, consequently some of the heterogeneity was

likely attributable to differences among study areas). Non-Euclidean models provided an opportunity to model one source of heterogeneity in an interpretable and explicit manner, thereby improving inference about abundance while reducing reliance on mixture distributions. Unlike prior applications of non-Euclidean SCR, most of Ontario's forests do not have obvious barriers or restrictions to bear movement (except in the south, where bear habitat is fragmented). This system, therefore, provides a valuable test of the applicability of non-Euclidean SCR models in situations other than where extreme landscape heterogeneity or habitat specialism leads to strong and predictable effects of structure on movement.

Our main objective was to investigate whether SCR models with non-Euclidean distances and models with two-point finite mixtures could improve inference about black bear abundance relative to Euclidean SCR models without mixtures. We predicted that if the covariates we chose to represent landscape structure had a meaningful influence on bear movement, then: (1) non-Euclidean models would improve fit relative to Euclidean models, (2) densities estimated from non-Euclidean models would be higher (less negatively biased) than densities estimated from Euclidean models without mixtures, and (3) non-Euclidean models would produce more precise estimates than Euclidean models with mixtures. Lastly, we assessed how robust non-Euclidean models were to variation in the spatial resolution at which landscape covariates were calculated.

## MATERIALS AND METHODS

### Study region

We monitored black bears in 51 study areas across Ontario in either 2017 or 2018 as part of ongoing monitoring (Fig. 1). The landscape we sampled covered 15,344.4 km$^2$; the areas covered by our arrays averaged 300.9 km$^2$ (Table S1). Most of our study areas (35/51) were in the Boreal Forest, which is a relatively unproductive habitat for black bears with a homogenous landscape dominated by coniferous tree species such as balsam fir (*Abies balsamea*), jack pine (*Pinus banksiana*), tamarack (*Larix laricina*), black & white spruce (*Picea mariana* and *Picea glauca*), and eastern white cedar (*Thuja occidentalis*; *Rowe, 1972*). Periodic logging and wildfires create a mosaic of different successional stages. The remaining 16 study areas were in the Great Lakes-St Lawrence Forest (GLSL) transition zone between the boreal and deciduous forest (*Boucher et al., 2009*). These forests are a mix of hard and softwoods dominated by white pine (*Pinus strobus*), red pine (*Pinus resinosa*), hemlock (*Tsuga canadensis*), American beech (*Fagus grandifolia*), yellow birch (*Betula alleghaniensis*), and sugar maple (*Acer saccharum*) (*Rowe, 1972*). The GLSL forest is a more productive habitat for black bears. It supports higher bear densities (*Howe, Obbard & Kyle, 2013*), but human population densities and harvest pressure are also higher, and busy roads and urban areas are more common, than in the Boreal Forest.

### Survey design and field sampling

In each study area, we installed 40 to 45 noninvasive stations designed to sample hair from black bears in curvilinear, branching arrays along secondary and tertiary roads or trails (stations were >30 m from any road), for a total of 2091 stations. Prior surveys of the same population used 20–25 traps at 2 km spacing; some study areas yielded insufficient

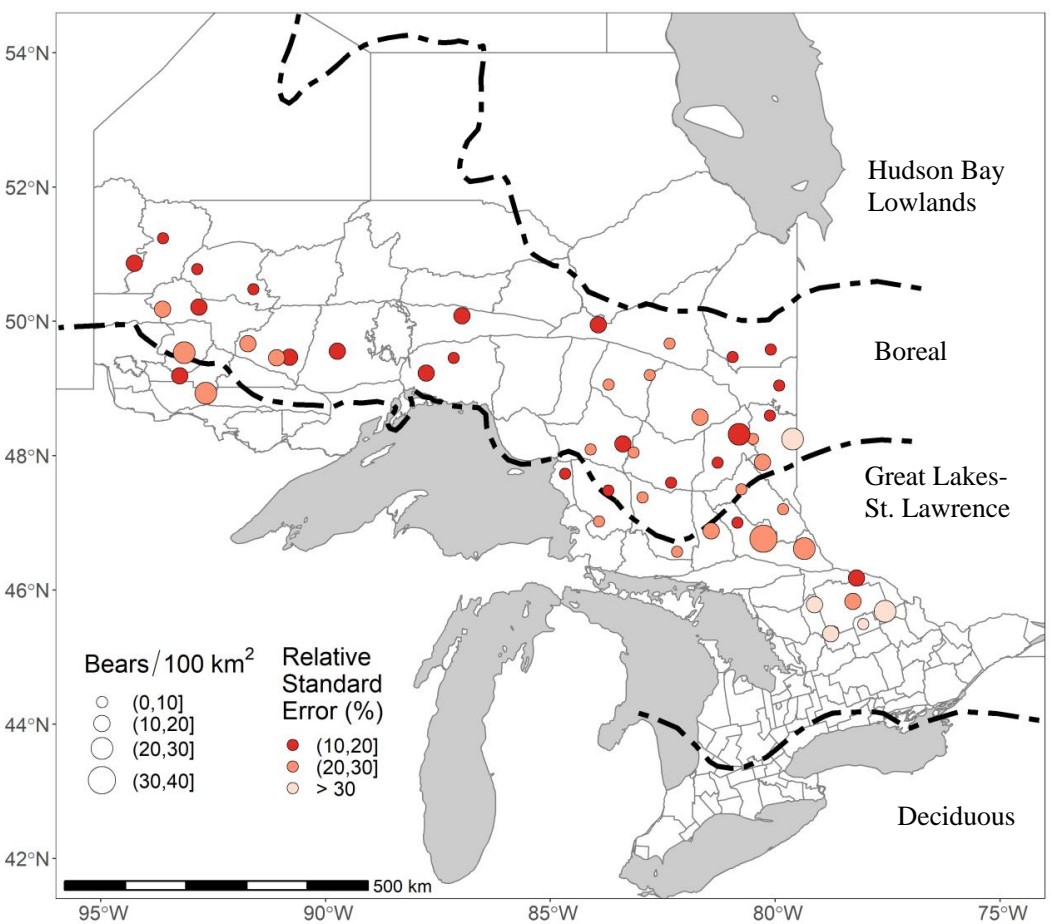

**Figure 1** **The 51 study areas in Ontario, Canada, where we estimated black bear density for study areas sampled in 2017–2018.** In this figure, bear densities were estimated with standard Euclidean spatially explicit capture-recapture models where $\sigma$ was a function of sex and $g_0$ was a function of sex and a site-specific-learned response to the detectors. RSE is the Relative Standard Error of the density estimate. Forest region boundaries (*Rowe, 1972*) are demarcated by dashed lines.

data to fit biologically realistic models, and estimates lacked precision (*Howe, Obbard & Kyle, 2013*; *Obbard, Howe & Kyle, 2010*). Since those surveys were designed, simulations and testing showed that optimal trap spacing varies with detectability at the trap location and that where detectability is low, as it is for black bears at hair corrals, optimal trap spacing is similar to or less than the expected scale of movements (" $\sigma$ " in the data analysis section below, *Clark, 2019*; *Efford & Boulanger, 2019*; *Sun, Fuller & Royle, 2014*; *Wilton et al., 2014*; *Sollmann, Gardner & Belant, 2012*). Applications to black bears showed that estimates of $\sigma$ for females were consistently >1 km, shorter in the southern US than in other environments, and usually around 2 km in Canada and the northern US; estimates for males were consistently >2 km and usually >3 km (*Hooker et al., 2020*; *Clark, 2019*; *Humm et al., 2017*; *Sun et al., 2017*; *Wilton et al., 2014*; *Howe, Obbard & Kyle, 2013*; *Sollmann, Gardner & Belant, 2012*; *Gardner et al., 2010*; *Obbard, Howe & Kyle, 2010*). We, therefore, increased

sampling effort and intensity for 2017–2018 sampling relative to previous work, spacing traps 1.5 km apart (slightly less than the expected scale of female movements) to provide larger samples of recaptures and spatial recaptures in the hope of improving inference about bear density.

At each station, we installed a barbed-wire corral to snag bear hair (*Woods et al., 1999*). We attached a single strand of barbed-wire approximately 50 centimeters above the ground. We assumed this height would avoid (or minimize) collecting hair from animals <two years old because morphometric data from live-trapping studies in Ontario showed that very few yearling bears reached 50 cm at shoulder height (*Obbard et al., 2017*; *Obbard & Howe, 2008*). Thus, we consider our density estimates specific to bears aged ≥ 2 years. We baited each corral with three partially opened tins of sardines in oil suspended at least 2 metres horizontal distance from any point along the barbed wire by a string from a board nailed approximately 2.5 m high on a central tree. We rebaited corrals and collected hair samples weekly from late May through early July for a total of 5 sampling occasions. If corrals were destroyed or became inaccessible, we attempted to replace them the same or the following week. We placed each sample, consisting of all the hairs on a single barb, into its own paper envelope, which we air dried and stored at room temperature until DNA extraction.

## DNA extraction and analysis

We attempted to genotype all samples with at least five hairs. We genotyped all extracted samples at 15 microsatellite loci and one sex-specific locus (see *Pelletier et al., 2012*). We discarded samples with >14 missing alleles. Also, when there were more than two alleles at a locus, the sample likely included DNA from more than one bear, therefore, we removed samples that had more than two alleles present at more than two loci. We then grouped the remaining samples into individual genotypes using 'allelematch' (*Galpern et al., 2012*). We set the number of allelic mismatches allowed between genotypes as 6 to 11. We checked all mismatches; if we could confirm the genotyping error, we corrected the genotypes; otherwise, we discarded the sample. Individuals represented by single samples required additional criteria for inclusion. We reamplified ambiguous samples, and if we could not verify them, we discarded the sample. We then used the unique genotypes to generate capture histories (For details see Appendix S2 in *Howe et al., 2022*).

## Data analysis

We conducted separate, independent SCR analyses of data from each study area. Raw movement data (i.e., distances between recaptures of individual) suggested that even male bears rarely travelled as far as 15 km between traps in the same study area, suggesting that home range diameters rarely exceeded this value. Consequently, we extended the regions of integration for each study area 15 km around all traps to ensure that bears with activity centers outside this region (*i.e.,* with activity centers >their home range radius from any trap) had very low to negligible probabilities of being detected (*Borchers & Efford, 2008*). We discretized the regions of integration to meshes of points (at 1 km spacing) that represented the possible activity center locations of individuals. We used

spatial data depicting waterbodies to exclude points that fell in lakes. We used functions implemented in the 'secr' R package (*Efford, 2020*) to ensure that this extent and resolution were adequate, *i.e.,* that density estimates were insensitive to increased extent or resolution for all data sets.

We considered five candidate models of a half-normal spatial detection probability function. All models included differences in both $g_0$ (detection probability at the activity center) and $\sigma$ (the scale parameter for the half normal detection probability function, which describes the decline in capture probability as a trap is placed further from the activity center) between sexes. An effect of prior detection at the same trap ($bk$) affecting $g_0$ was also included in all models. These forms of variation in detectability are frequently supported in SCR data from black bears sampled at baited barbed-wire corrals (*Hooker et al., 2020*; *Azad et al., 2019*; *Howe, Obbard & Kyle, 2013*; *Gardner et al., 2010*). The simplest model included only these effects (sex-specific $g_0$ and $\sigma$ and the $bk$ effect on $g_0$). A second model also included an additive effect of individual heterogeneity modelled as a two-point finite mixture distribution ($h2$) affecting $\sigma$. In these two models, detection probability declined with the Euclidean distance between activity centers and traps. In the remaining three models, detection probability declined with increasing non-Euclidean distance as described below, and the mixture distribution was omitted.

Variable survey effort among sampling occasions caused by, for example, destruction of traps by falling trees, road washouts, or changes to work schedules, was explicitly accounted for in the SCR data (*Efford, Borchers & Mowat, 2013*). We fit the models by maximizing the conditional likelihood for proximity detectors and calculated bear density as a derived parameter from each model fitted to data from each study area. We estimated asymptotic variances assuming an underlying homogeneous Poisson point process for the distribution of activity centers (*Efford, Borchers & Byrom, 2009*; *Borchers & Efford, 2008*). We used R version 3.6.0 (*R Core Team, 2020*) and version 4.2.0 of the 'secr' package (*Efford, 2020*).

To fit non-Euclidean models, we followed the procedures in *Sutherland, Fuller & Royle (2015)* to dynamically transform the matrix that describes distances between each detector and each point in the region of integration (see Data S1 for an example). We defined the non-Euclidean distance as the accumulated cost of the least-cost path from activity centers to traps with the *Dijkstra (1959)* algorithm implemented in the 'gdistance' R package (*Van Etten, 2017*). These paths were calculated from a cost surface that was a digital representation of a landscape covariate that was additionally transformed by an $\alpha_2$ parameter depending on whether the covariate impedes ($\alpha_2 < 0$) or facilitates ($\alpha_2 > 0$) movement. We used estimates from the simplest model as starting values when fitting models with mixtures or non-Euclidean distances.

## Movement covariates

We chose a set of covariates for the non-euclidean model by reviewing the literature and based on our knowledge of black bears that we hypothesized related to the cost or ease of movement. Roads are generally an obstacle to wildlife (*Bennett, 2017*), and high-traffic roads are a high-risk obstacle for black bears (*Ditmer et al., 2018*; *Karelus et al., 2017*; *McCown et al., 2009*; *Beringer, Seibert & Pelton, 1990*; *Rogers & Allen, 1987*). However, black bears

also use roads to travel (*Witmer, 2019*; *Lewis et al., 2011*; *Manville, 1983*). Traffic intensity and size of the road or even the density of roads is an important factor that promotes or disrupts black bear movement (*Ditmer et al., 2018*). We calculated the areas occupied by linear features such as highways, roads, and resource roads from spatial data describing the road network of Ontario (*OMNRF, 2015a*; *OMNRF, 2015b*). We first rasterized all roads to 10-meter resolution and assigned all pixels that had roads a value of "1" and all other pixels a value of "0". This allowed us to remove overlapping road segments between datasets. For this movement covariate, we allowed the $\alpha_2$ parameter to explore both the positive and negative parameter space. A negative value indicated that linear features facilitate black bear movement, while a positive value indicated that roads impede movement.

Moving through rugged terrain requires more energy than travelling across flat terrain. We calculated the Vector Ruggedness Measure, a proxy for landscape ruggedness (VRM; *Sappington, Longshore & Thompson, 2007*), from a 30-meter resolution Digital Elevation Model (*OMNRF, 2015c*). We used the ArcGIS Benthic Terrain Modeler to calculate Vector Ruggedness Measure with a neighborhood size of 3 (*Wright et al., 2012*). We estimated the $\alpha_2$ parameter for this movement covariate on the log scale since a negative value would indicate that rugged terrain facilitates black bear movement.

Lastly, there was scant evidence of black bears crossing large waterbodies (but see *Rogers & Allen, 1987*), and we expected that travelling through water would require bears to expend more energy than when travelling overland. Consequently, across our study areas large waterbodies acted as barriers to bear movement, and smaller waterbodies hindered movement. We first quantified the cost of travelling through areas with water. We calculated the area occupied by waterbodies in a 15-meter resolution land cover map (*OMNRF, 2014*). We combined the "clear open water" and "turbid water" classes, to which we assigned a value of "1" with all other classes assigned a value of "0". We also estimated the $\alpha_2$ parameter for this movement covariate on the log scale since a negative value would indicate that waterbodies facilitate black bear movement.

We aligned all raster surfaces to the terrain ruggedness layer. We then coarsened the water and road layers to match the resolution of the terrain ruggedness layer (30 m). We aggregated the water layer by a factor of two using a sum; this gave us a range of values from zero to four, where we defined a value of one as 225 m$^2$ of water, and a value of four as 900 m$^2$ of water. We aggregated the road layer by a factor of three using a sum, yielding values from zero to nine, where a value of one was defined as 100 m$^2$ of roads, and a value of nine was 900 m$^2$ of roads. We then scaled the values of each of these surfaces between 1–10 so that parameter estimates from all landscape covariates were comparable. We did not allow for zeros in the cost surfaces since cost values >0 are necessary to compute the least-cost paths. Finally, we created boundary layers for each study area by buffering 15 kilometers (matching the region of integration) from each detector and clipped all three spatial layers using this boundary. We projected all layers to NAD83 / Ontario MNR Lambert (EPSG:3161).

## Spatial resolution

To assess the influence of the spatial resolution at which costs, and therefore least-cost paths, were calculated on our results, we spatially aggregated our data into coarser resolutions. Our finest scale resolution was 30-meters. We aggregated movement covariates to resolutions of 60, 120, 240, 520 and 960 m using the mean value across cells and refit the three non-Euclidean models previously described. Therefore, we fit a total of 20 SCR models to each data set: two Euclidean models and 18 non-Euclidean models (three covariates calculated at each of six resolutions).

## Model and estimate comparisons

For each study area and spatial resolution, we compared the goodness of fit of complex models (*i.e.*, finite mixture or non-Euclidean distance models) to the simplest model (the Euclidean distance model without mixtures) using a likelihood-ratio test with an $\alpha$ level of 0.05. We did not use model selection criteria (*e.g.*, Akaike's Information Criterion) because we were not interested in minimizing the bias–variance tradeoff among a set of models, which is the purpose of statistical regularization techniques such as model selection (*Hooten & Hobbs, 2015*). Rather, we were interested in understanding whether elements of the more complex models were redundant. We then summarized the change in density and the associated change in relative standard error (RSE; calculated as the standard error divided by the point estimate) when a more complex model significantly improved fit. We also investigated changes in log-likelihood, density and RSE between Euclidean finite mixture and non-Euclidean models that both significantly improved fit relative to their simpler Euclidean model counterparts. We filtered out models with density estimates over one bear per km$^2$ since densities this high in Canada or the northern USA are unrealistic and have not been reported (*Welfelt, Beausoleil & Wielgus, 2019*; *Roy et al., 2012*; *Gardner et al., 2010*; *Miller et al., 1997*). We also filtered out models that had RSEs of density >40% because they were too imprecise to inform management. Due to the large number of models and computational requirements, we fit all models using a serial farm on several computer clusters (computecanada.ca RRG gme-665-ab).

## RESULTS

A total of 34,235 hair samples were collected, 11.3% were not processed because the envelope did not contain enough hairs, 9.5% were excluded because DNA amplification or genotyping failed, 4.4% were mixed samples from more than 1 bear, 0.2% were excluded because they were unique genotypes or the sample was mislabeled, and 25,534 (74.6%) were successfully genotyped and assigned to an individual (Table S2). There were no obvious spatial or temporal patterns in genotyping success rates. We identified 2,534 unique bears with a total of 8,259 independent detections. The median number of detections of bears in a study area was 145 and ranged from 38 to 350; the median number of individuals was 49 and ranged from 23 to 82. The median number of detections per individual ranged from 1.6 to 5.2 across study area. The median number of recaptures per study area was 100 and ranged from 15 to 283. The median number of spatial recaptures was 83 and ranged from 10 to 254 (Table S2).

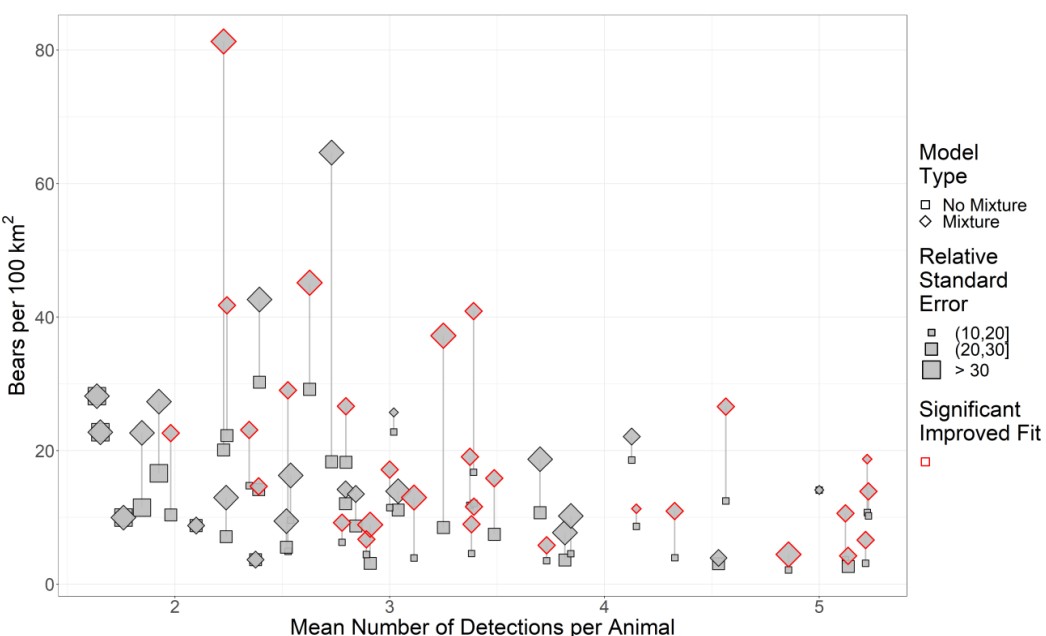

**Figure 2** **The effect of sample size and two-point mixture models on black bear density estimates and their associated Relative Standard Error (RSE).** We estimated black bear density for 51 study areas sampled in 2017–2018 in Ontario, Canada. Significance was calculated using a likelihood-ratio test between the base model without mixture (null model) against the more complex model with mixture. Symbols in red are mixture models that significantly improved fit. Grey lines indicate pairs of models, and the length of the line is the change in the density estimate.

Across our study areas, RSEs of density estimates generally increased as the sample size (as the mean number of detections per animal) decreased (Fig. 2). Consequently, we removed 26 study areas from our analysis since study areas with detections per animal (DPA) ≤ 3 were already problematic for mixture models (Fig. 2).

## Finite mixtures and Euclidean models

For the 25 study areas with DPA >3, we compared mixture models to their simpler counterparts using likelihood-ratio tests. We found that when we included a two-point mixture, it significantly improved fit in 17 out of 25 cases (Table 1). One of these models that outperformed the simpler model had an RSE >40%. The 16 remaining mixture models yielded density estimates that were, on average, twice as high (mean increase = 100.4%), with RSEs that were, on average, 44.7% larger, relative to estimates from the simple model fit to data from the same study area. We reviewed estimates of the proportions of individuals allocated to the different latent groups and of sex-specific $\sigma$ for both groups and found that in 11 of 16 cases, >64.1% of individuals were assigned to the group with lower $\sigma$ (>80.1% in 7 cases) and estimates of $\sigma$ for this group was lower than expected for the species and sex (<1 km for females or <2 km for males). These 11 mixture models yielded estimates that were 124.8% higher, with RSEs that were 51.9% larger relative to estimates from the simpler model.

**Table 1  Summary statistics for models that significantly improved fit over standard Euclidean models without a two-point mixture.** These models were used to estimate black bear density across 25 study areas in Ontario, Canada, for 2017–2018. The average change in density and Relative Standard Error (RSE) was only measured for models that significantly improved fit over their Euclidean counterparts without mixture. Significance was calculated using a likelihood-ratio test between the base Euclidean model without mixture (null model) against the more complex models.

| | Euclidean mixture models | Non-Euclidean models with 120-meter pixel resolution | | |
| --- | --- | --- | --- | --- |
| | | Road density | Terrain ruggedness | Waterbody density |
| Number of models | 17 | 12 | 3 | 3 |
| Number of models with density >1 bear per km$^2$ | 0 | 2 | 0 | 0 |
| Number of models with RSE >40% | 1 | 4 | 0 | 0 |
| Remaining after filtering | 16 | 8 | 3 | 3 |
| Average $\Delta$ Density % | 100.4 | 65.6 | −53.0 | 30.2 |
| Average $\Delta$ RSE % | 44.7 | 29.8 | −10.0 | 46.7 |

## Non-Euclidean models

There was no general pattern in the performance of non-Euclidean models relative to Euclidean models across spatial resolutions (Table 2). We found general agreement across spatial resolution for the terrain ruggedness and waterbody density models, but they were supported in a few study areas. In contrast, we found that road density models were more often supported at a spatial resolution of 240-meters.

Non-Euclidean model support varied substantially depending on spatial resolution; therefore, to provide a more detailed overview of our results, we discuss the patterns found in models that were fit with spatial data at 120-meter resolution. At this spatial resolution, there is a reasonable tradeoff between biological realism (*i.e.,* the landscape is, in fact, continuous) and computation time (<40 h Fig. S1). When we increased complexity from Euclidean to non-Euclidean models, at least one of our three non-Euclidean models significantly improved fit to data from 15 of 25 study areas. Modelling distance as a function of road density improved fit in 12 cases, but 4 of these yielded unreasonable density estimates of over one bear per km$^2$ or RSE >40% (Table 1). The remaining eight models yielded higher density estimates than simpler counterpart models, but the increase (65.6%) was lower than for two-point mixtures. RSE also increased less than when we modelled heterogeneity using finite mixtures. Terrain ruggedness and waterbody density significantly improved fit to data for only three different study areas (Table 1). Density estimates for terrain ruggedness models decreased relative to simpler models whereas those from waterbody density models, increased; RSE decreased for terrain ruggedness and increased for waterbody density.

We compared the log-likelihood, density estimates, and relative standard error of each Euclidean two-point finite mixture model and each non-Euclidean model without mixtures that significantly improved fit and had a reasonable density estimate ($\leq$ 1 bears per km$^2$) and RSE ($\leq$ 40%) over their simpler counterpart (Table 3). Both types of complex models

**Table 2  The number of non-Euclidean models that significantly improved fit over standard Euclidean models without two-point mixture by pixel resolution.** These models were used to estimate black bear density across 25 study areas in Ontario, Canada, for 2017–2018 for 6 different pixel resolutions.

| Movement Covariate | Pixel resolution in meters | | | | | |
|---|---|---|---|---|---|---|
| | 30 | 60 | 120 | 240 | 480 | 960 |
| Road density | 8 | 8 | 12 | 15 | 12 | 8 |
| Terrain ruggedness | 3 | 3 | 3 | 3 | 3 | 1 |
| Waterbody density | 2 | 2 | 3 | 2 | 3 | 2 |
| Sum | 13 | 13 | 18 | 20 | 18 | 11 |

significantly improved fit to data from seven study areas. A total of nine non-Euclidean models improved fit to data from these study areas (four road density models, three terrain ruggedness models, and two waterbody density models; Table 3). The non-Euclidean road density models yielded higher density estimates and slightly lower RSE than Euclidean mixture models, whereas non-Euclidean terrain ruggedness models yielded lower density estimates and RSE. Waterbody density models had lower density estimates but higher RSE.

We estimated landscape resistance induced by movement covariates. We found that there were three study areas where roads impeded movement, and there were five where roads facilitated movement (Fig. 3A). We could not confidently associate this pattern to a difference in traffic volume or intensity of human impact since the southern study areas where high-traffic roads were most prevalent were removed from the analysis due to the low sample size. For terrain ruggedness, some estimates of $\alpha_2$ were relatively high, and in three cases, they significantly improved fit over their Euclidean counterpart (Fig. 3B). One of these study areas was found in a relatively rugged area in Ontario, but the two others were found in areas where terrain ruggedness was not substantially high compared to all the other study areas. Generally, these terrain ruggedness models were driven by a few clusters of rugged terrain that isolated a single detector. There were three study areas where waterbody density models significantly improved fit over their Euclidean counterparts, and the associated $\alpha_2$ estimates were generally high compared to the other study areas (Fig. 3C). We did not find that these study areas had relatively high or low waterbody density. However, waterbodies did separate stations, and their configuration isolated a series of stations.

## DISCUSSION

We predicted that non-Euclidean SCR models would account for otherwise unmodelled detection heterogeneity; consequently, these models would improve fit relative to conventional Euclidean models even where no dramatic landscape features obviously influenced movement. Further, we predicted that non-Euclidean models would yield more precise estimates than the more commonly used finite mixture methods for modelling individual heterogeneity due to unobserved sources because they capture meaningful, observable biological patterns related to movement. We found that 26 of 51 data sets could not support the additional complexity required to model individual heterogeneity without
**Table 3 Summary statistics for the difference between non-Euclidean models and Euclidean models with a two-point mixture that significantly improved fit over standard Euclidean models without a two-point mixture.** These summaries were for black bear density estimated from seven study areas in Ontario, Canada, for 2017–2018. The average change was measured as the average percent difference in log-likelihood (LL), density, and Relative Standard Error (RSE) between non-Euclidean models with 120-meter pixel resolution relative to Euclidean models with a two-point mixture that significantly improved fit over Euclidean models without a two-point mixture.

|  | Road density | Terrain ruggedness | Waterbody density |
|---|---|---|---|
| **Number of models compared** | 4 | 3 | 2 |
| **Average Δ log-likelihood %** | −0.7 | −0.2 | −1.2 |
| **Average Δ Density %** | 91.9 | −75.3 | −48.9 |
| **Average Δ RSE %** | −0.5 | −44.5 | 33.0 |

producing unrealistic estimates with high associated uncertainty. Even in larger data sets (those with more than three detections per animal on average), results of modelling individual heterogeneity on density estimates was highly variable among methods (mixtures or non-Euclidean distances), landscape covariates, pixel resolutions, and study areas. Furthermore, the precision of estimates from non-Euclidean models was poor, like that from two-point mixture models.

Contrary to past applications of non-Euclidean models where a single landscape feature or characteristic had strong and consistent effects on animal movement (*Sharma et al., 2020*; *Sutherland et al., 2018*; *Tobler et al., 2018*; *Morin et al., 2017*; *Fuller et al., 2016*; *Murphy et al., 2016*; *Sutherland, Fuller & Royle, 2015*), our results suggest caution must be applied when using non-Euclidean models in landscapes and for species with less clear predictors of movement. We did find that, in some cases, black bear detection heterogeneity was explained by landscape heterogeneity, but for less mobile species and species with low detection, these complex models should be used with caution. Particular caution is required when using model selection approaches such as the Akaike information criterion because it is more liberal in including variables compared to likelihood ratio tests (*Murtaugh, 2014*). The most obvious impediments to bear movements in Ontario are areas of human development, including large busy roads (also see *Morin et al., 2017*). We hoped that non-Euclidean models that considered road density would reveal patterns, showing resistance only where roads were busier and occurred at higher densities. Unfortunately, in areas of high human population density where black bear movement is restricted and anthropogenic foods are often available, bears visited fewer traps such that sample sizes and, therefore, power to detect effects of landscape structure was reduced. Consequently, we did not acquire enough data to successfully fit non-Euclidean models in the same areas where we expected the strongest effects of landscape structure.

Our surveys were not designed to detect landscape resistance, and they could not have been realistically designed for this type of analysis because we considered multiple possible landscape covariates with different spatial patterns. Furthermore, although we obtained reasonably large sample sizes (detections, recaptures, and spatial recaptures), our curvilinear arrays provided fewer opportunities to recapture animals at different locations compared to a regular grid with the same trap spacing. In addition, we collected data

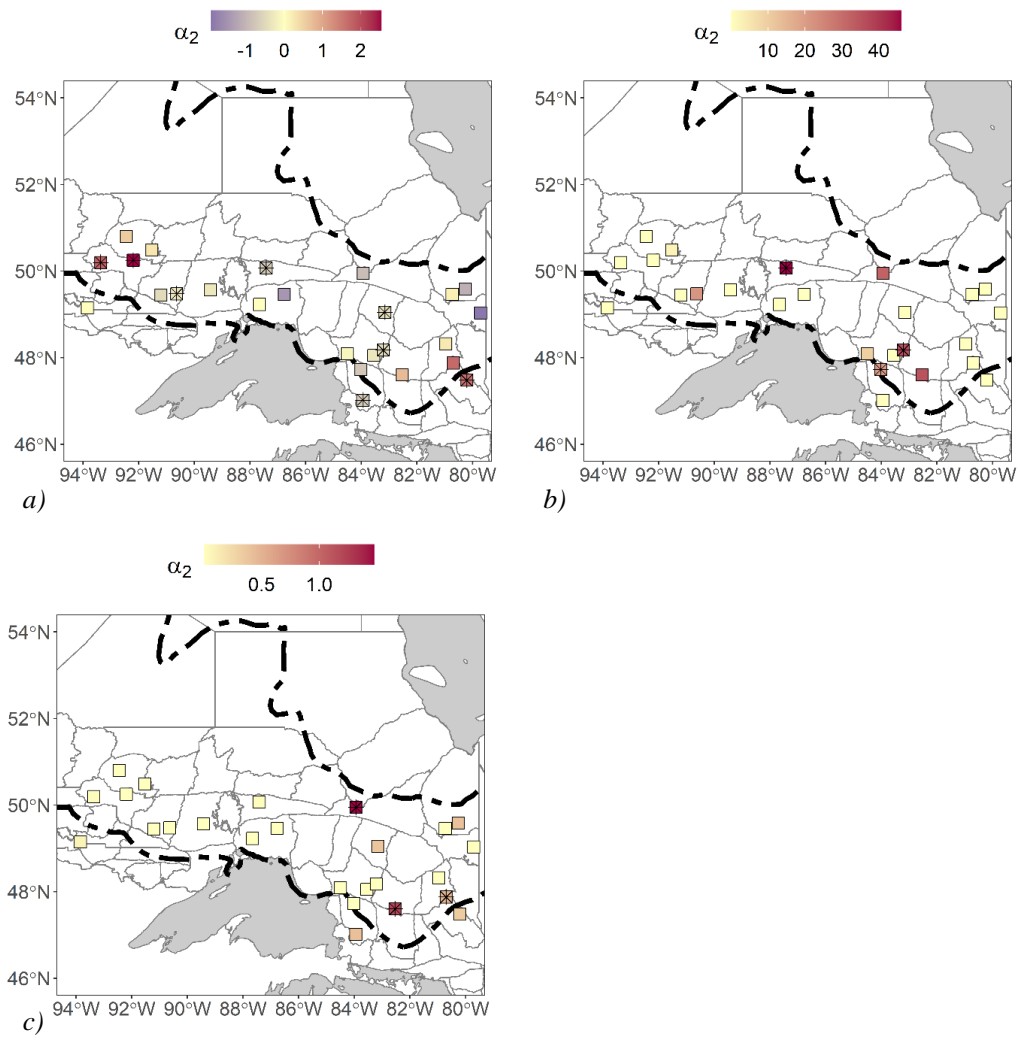

**Figure 3  Black bear non-Euclidean $a_2$ parameter estimates for 25 different study areas.** Study areas were in Ontario, Canada and hair samples were collected between 2017–2018. We generated these non-Euclidean models from spatial data with a pixel resolution of 120 m. (A) road density, (B) terrain ruggedness, and (C) waterbody density. Study areas with "*" were models that significantly improved fit over their Euclidean counterparts using a likelihood-ratio test. Forest region boundaries (*Rowe, 1972*) are demarcated dashed lines.

on only five occasions; as a result, opportunities for recaptures and spatial recaptures were limited. More intensive surveys might yield sufficient data to detect and effectively model the effects of landscape structure on movements even outside of highly structured environments. Pooling our data across study areas could provide greater power to detect and model different sources of heterogeneity (*Proffitt et al., 2020*; *Morin et al., 2018*; *Howe, Obbard & Kyle, 2013*), including landscape resistance, but we opted against this for several reasons. First, we preferred to emphasize replication because many other surveys are comparable in size and effort to our study-area specific surveys (*i.e.,* a single array of 40–45 detectors), but relatively few researchers could reproduce what we might be able to achieve

with our complete data set, which is part of a government funded monitoring program. Second, detectability varies even among study areas in similar habitats (*Howe et al., 2022*); therefore, we were concerned that pooling data would induce additional heterogeneity in detectability which would also need to be modelled (*Moqanaki et al., 2021*). Third, we suspected that road density could have qualitatively, different effects on different study areas; consequently, pooling data could obscure rather than reveal the effects of roads. Finally, fitting non-Euclidean SCR models is computationally intensive; fitting them to data pooled across study areas could become prohibitively time-consuming and demanding of computational resources.

When simple SCR models are fit to heterogeneous data, there is a risk of reporting negatively biased estimates of abundance and underestimating the uncertainty associated with those estimates. Modelling heterogeneity whenever possible is therefore recommended. However, the shortcomings of finite mixture models and the need to apply them with care are well known (*Pledger & Phillpot, 2008*). Although they can reduce negative bias due to unmodelled detection heterogeneity, they have stringent data requirements and may yield estimates of similar accuracy but reduced precision, especially when fit to sparse data (*Link, 2004*; *Link, 2003*; *Pledger, 2000*). Here, there was no consistent pattern in the magnitude of increase in density estimates when more complex models were used (consistent with sensitivity of abundance estimates to the parameters of the mixture distribution). Where finite mixture models significantly improved fit, density estimates were twice as high as those estimated from models that explicitly accounted for heterogeneity attributable to sex, prior detection, and spatially variable exposure to traps, in an analysis that minimized the potential for spatial heterogeneity (*Moqanaki et al., 2021*) by analyzing study area-specific data. Furthermore, in 11 of 16 cases where mixture models significantly improved fit and yielded biologically plausible estimates of density, estimates of $\sigma$ for different sexes and latent groups indicated that most individuals with activity centers within 2–5 km of our arrays were likely to go undetected because they were unlikely to travel that far. Further, the estimates of $\sigma$ equate to biologically implausible areas of use for this species. We suggest that these inferences are potentially flawed, and possibly an artifact of limited opportunities to detect animals and the fact that both activity center locations and the proportions of animals in different groups are modelled as latent effects. The risks associated with providing inflated and uncertain estimates of animal abundance to decision-makers are potentially severe, including overharvest and the failure to provide protection when and where it is needed. Therefore, we recommend that practitioners use finite mixture models cautiously and inspect estimates of all model parameters for biological realism before drawing inferences.

## CONCLUSIONS

Accounting for individual heterogeneity in the scale of animal movements is critically important when estimating animal abundance or density using SCR models. Developing and testing methods for appropriately and mechanistically capturing this heterogeneity is thus an important topic for basic and applied research programs aiming to use these

methods. Non-Euclidean SCR models are a tantalizing advancement because we know that animals do not move randomly throughout study areas and indeed have highly structured movements that are influenced by the landscape. Our results suggest caution in applying these models when there is not a large sample of spatial recaptures (total and per individual) or when there is not a very clear landscape covariate that is known to influence the movement of the focal species. However, caution is nuanced and likely dependent on the spatial arrangement of the detectors. Studies combining GPS-telemetry and SCR data could allow us to better understand how animals move through landscapes and how this movement is captured in SCR data. Finally, individual random effect models show promise for accounting for individual heterogeneity in CR models (*White & Cooch, 2017*); the next step would be to incorporate this approach into SCR models.

## ACKNOWLEDGEMENTS

Field surveys were conducted by the Ontario Ministry of Natural Resources and Forestry's Regional Operations Division. The authors would like to thank field crews, field coordinators (Darren Elder, Emilie Kissler, Gillian Holloway, April Mitchell, Jay Fitzsimmons, Lyle Walton, and Tim Cano), Katelyn Jackson, Emily Walker, Jade Black, Sarah Langer, Linda Rutledge, and many staff and students for assistance in the genetics lab, Kelly Lauder, Ian Petreman, and Norm Mooyekind for data management solutions, and Kevin Middel, Tore Buchanan, Peter Carter, Bob Watt, Martyn Obbard, and Erica Newton for their support during various stages of the project. We also would like to thank Dr. Pierre Dupont and 4 anonymous reviewers for reviewing earlier versions of this manuscript.

### Funding

This work was supported by the Natural Sciences and Engineering Research Council of Canada Discovery Grant to Joseph M. Northrup. Also, this research was enabled by support provided by Compute Canada (RRG gme-665-ab; www.computecanada.ca). The funders had no role in study design, data collection and analysis, decision to publish, or preparation of the manuscript.

### Grant Disclosures

The following grant information was disclosed by the authors:
The Natural Sciences and Engineering Research Council of Canada Discovery Grant.
Compute Canada (RRG gme-665-ab; www.computecanada.ca).

### Competing Interests

The authors declare there are no competing interests.

### Author Contributions

- Robby R. Marrotte conceived and designed the experiments, performed the experiments, analyzed the data, prepared figures and/or tables, authored or reviewed drafts of the article, and approved the final draft.

- Eric J. Howe conceived and designed the experiments, authored or reviewed drafts of the article, designed the surveys, and approved the final draft.
- Kaela B. Beauclerc conceived and designed the experiments, authored or reviewed drafts of the article, coordinated and conducted the lab work, and approved the final draft.
- Derek Potter conceived and designed the experiments, authored or reviewed drafts of the article, designed the surveys, coordinated the field surveys, and approved the final draft.
- Joseph M. Northrup conceived and designed the experiments, authored or reviewed drafts of the article, and approved the final draft.

### Field Study Permissions

The following information was supplied relating to field study approvals (i.e., approving body and any reference numbers):

Ontario Ministry of Northern Development, Mines, Natural Resources and Forestry.

### Data Availability

The data we used for this analysis were a subset of a much larger dataset available on Dryad: Howe, Eric (2021), Spatially explicit genetic capture-recapture data from black bears in Ontario, Canada, 2017-2019, Dryad, Dataset, https://doi.org/10.5061/dryad.7wm37pvtz.

### Supplemental Information

Supplemental information for this article can be found online at http://dx.doi.org/10.7717/peerj.13490#supplemental-information.

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
