# Peer review of "Explaining detection heterogeneity with finite mixture and non-Euclidean movement in spatially explicit capture-recapture models"

_PeerJ, doi:10.7717/peerj.13490_

## Round 0.1 · original submission · Major Revisions

Dear Dr. Marrote,

After the review by three reviewers, your manuscript may be accepted in PeerJ after some improvements are addressed in your manuscript. I acknowledge the role played by the reviewers, who suggested significant changes for the improvement of your manuscript. Please find the suggestion attached to this letter.

Sincerely,
Daniel Silva

Reviewer 1 ·

Basic reporting

no comment

Experimental design

no comment

Validity of the findings

no comment

Additional comments

The paper “Explaining detection heterogeneity with finite mixture and non-Euclidean movement models” by Marrotte et al. is an analysis of finite mixture, non-mixture, and non-Euclidean distance (non-ED) models to address individual heterogeneity in detection rates of black bears in Ontario. The paper is well written and clear and the analysis methods seem sound. Below are my line-by-line comments followed by a summary.

L1. Perhaps include something in the title to indicate that you are talking about spatially explicit CMR.
L18. What do you mean by “recently”? Do you mean that others have recently done this or that you have recently done this? Please be more specific.
L28. Wide CIs are not necessarily a bad thing if warranted. Excessive individual heterogeneity or estimation issues could make the results unreliable but you would want to know that wouldn’t you? The worst possible scenario is to have an erroneous estimate with narrow CIs.
L32. The fact that there was no rhyme or reason for why certain resolutions were supported in one study and not in another is a possible red flag. Sparse data could cause this. After all, if you test enough covariates, something is going to be significant by chance alone (about 1 in 20 if alpha=0.05).
L33. Yes, the finite mixture models are a black box and finicky. However, the wide CIs (and I assume the high point estimates had wide CIs too) is not a good measure of model validity (unless of course they are ridiculously wide). Wide CIs can be a warning that the point estimate should be approached with caution.
L53. Prior exposure to sampling devices constitutes a behavioral effect (trap happy/shy) and is modeled differently from individual capture heterogeneity. I would take that one out of the list.
L101. Or that the features are similar for every animal.
L216. The lake sentence seems out of place.
L227. The sentence seems to refer to only 2 models not 3. Please clarify.
L245. Please provide a little more information in the text. Readers should not have to read another paper to understand what you did. Were these α2 parameters set by you or estimated from the data?
L263. What do you mean by “let”? Was this an option in the algorithm?
L289. Do you mean water?
L313. Finite mixtures account for more than just spatial variation. Are all these covariates on g0 or on σ as well?
L320. Did you scale the variables to facilitate convergence? The distance variables can be troublesome if not scaled in some way.
L322. Did you consider combining some of your study areas with similar habitat? You could still analyze them as separate sessions in secr but you might allow for some parameter sharing across areas. Home range sizes and capture probabilities likely do not vary much among your study areas with similar habitat. The extra data and the sharing of parameters across study areas could really help refine your estimates.
L337. Maybe this model is telling you not to rely too much on the estimate…
L374. Troubling that an effect due to roads was not predictable. Pooling across study areas could help.
L393. Whereas the finite mixtures operate on other variables besides movement. A good thing…
L394. In total one of the largest ever, but the individual study areas were fairly small and that is the level that these estimates were derived.
L403. Explain what you mean here by comparability.
L407. Agreed; non-ED models seem very inconsistent.
L412. Yes, but low precision can be a good thing if estimates really are imprecise. Estimates that are deemed to be more precise than they really are can be more insidious. Also, what about plots of various levels of pmix to identify multimodality (https://www.otago.ac.nz/density/pdfs/secr-finitemixtures.pdf)?
L413. I do not understand this sentence.
L428. Good point.
L445. Do you mean run them all simultaneously with each study area a session?
L460. What about random effect (intercept) approaches as have been developed for non-spatial data? See White, G.C. and Cooch, E.G., 2017. Population abundance estimation with heterogeneous encounter probabilities using numerical integration. The Journal of Wildlife Management, 81(2), pp.322-336.
Table 2. The table title does not seem to go with the table.
Table 3. The difference in log likelihood does not seem very high for any of them. This looks like a lot of trouble for little benefit compared with finite mixtures.
Figure 2. Are the non-mixture models the null models or the non-ED models?

This is an important topic and this paper is a good start. As you know, the spatially explicit estimators already absorb a lot of individual capture variation in CMR models due to the animal’s location on the trapping grid. The non-ED models just refine the distance part of the detection function. I always felt like the mixture models in a spatially explicit context were capturing something else. With bears, there are always individuals that approach a hair trap but never go in. This may have nothing to do with movement but, instead, relate to the innate behavior of individual bears. Consequently, it makes sense to me that mixture models had higher N because non-ED models address only capture heterogeneity due to the location of the animal on the trapping grid. I think that this (heterogeneity not related to movement) effect is probably heightened in bears because of their high intelligence (and individualism). I like your suggestion to include movement data based on telemetry, but I think that you have overlooked opportunities for pooling data across individual study areas, at least some of them. Perhaps you could combine data from some of your more similar study areas to help reduce the noise in your study because of estimation problems. Then you might find more consistent relationships between non-ED variables and scale resolution. I would want to see clear relationships between the 3 model types, scale resolution, and variables describing costs of travelling. As it stands now, your study does not present much of a way forward.

Reviewer 2 ·

Basic reporting

There are several places where more description is necessary, outlined below in approximate order of importance.
L65 and L74: For the paragraphs starting on these lines, can you provide examples (rather than just citations) of studies that came to these conclusions and support your statements? The first of these 2 paragraphs could also be fleshed out more, or these two could be combined.
L93: Do you mean within SCR frameworks, or independent of whether it is SCR/CR?. These two sentences about finite mixture is awkward within this SCR paragraph.
L137: Check grammar: “models fit was”
L145: Can you briefly describe that the expected sources of individual heterogeneity where?
L200: What was the range in spatial extent across your 51 study areas?
L322: How many samples were discarded due to too many mismatches or missing genotypes?
L342: Why did you expect that performance of non-Euclidean models would increase at finer resolutions? This wasn’t been explained.
L445: Define or provide examples of ‘multistrata’. This term has not been previously used.
Table 2 legend needs more detail to stand alone.
Table 3 legend: Summary statistics are “for” or “of”, not between
Figure 1 caption does not say directly that densities are shown, and does not define RSE. Is this all 51 locations? Why is there “deciduous” in the bottom right corner of the figure if there aren’t any bear densities shown south of that last dotted line?

Experimental design

L384: Study sites for non-Euclidean models need to be located relative to the hypothesized resistance covariate such that the pattern of detections are actually informative. Were there pairs of sites between which the resistance was high, and then also pairs of sites where the resistance was low?
L250: What is the range of observed movement covariates across studies and landscapes? Also, see comment above.
L217: The description of your model set is hard to follow. For example, you describe that you had 5 models all with sex as a covariate on detection and sigma and behavior on detection. Then you go on to talk about 3 other models, but I do not know what differentiated those first 5 models. A table would be helpful.
L204: What does “two mixed loci” mean? More than three alleles at a locus?
L208: It is unclear when and which samples were discarded. Which sample of the mismatch pair did you discard, and was it the 6-11 mismatches threshold that was used to determine when a sample was ultimately kept or not?
L312: Provide the calculation for RSE.
L270, 281: How did you force/constrain an estimated parameter? I think you are referring to your log() transformation, in which case that line (L247) is better placed here.
L213: The term “integration mesh” is not common usage. Do you mean habitat mask or state-space? If you decide to continue using ‘integration mesh’, quickly define.

Validity of the findings

A landscape may not have extreme covariate values, but another reason for the lack of support for a movement covariate is sub-optimal trap placement. The methods did not describe this aspect of the sampling and therefore how powerful or appropriate the design was for non-Euclidean models, so it is difficult to assess these results. However, I do agree these more complex models require more recaptures and larger sample sizes.

Can you offer any explicit implications or repercussions for inference if a non-Euclidean model is used when it shouldn’t have been? Unrealistically large density estimates, impacts on RSE?

Additional comments

In this manuscript, the authors investigate the ability of non-Euclidean and finite mixture SCR models to improve on the basic SCR model by better accounting for heterogeneities induced by individuals and the landscape. They use a large dataset of over 50 black bear studies in Ontario. The concepts were generally well explained, and I have mainly only minor comments throughout the manuscripts about where clarification or more detail is needed. However, I do have two larger issues. The first is that they describe the non-Euclidean model as a way to account for detection heterogeneity, as evident in their title, but a more rigorous description is that it primarily serves to describe the pattern of movement and habitat use, which more accurately and precisely place activity centers on the landscape, and which also impacts the detection model. The authors never quite directly frame the problem as such and focus their discussion largely on detection. Secondly, the authors do not describe whether their study designs were appropriate for non-Euclidean models, and this was not addressed in the Discussion. The authors conclude that the more complex models are not always appropriate or possible to apply due to small sample sizes or weak landscape patterns (which I agree with), but failed to consider their sampling design. This lies at the crux of their conclusions. I believe that changes to the manuscript that address these and the other smaller points would improve the manuscript for publication in PeerJ.

Reviewer 3 ·

Basic reporting

This article has a colid presentation. I was very impressed by the breadth of literature references ranging from SCR-cost modeling to statistical nuances of model selection, etc. The authors are clearly well-read on these topics. The language is clear (though I note some areas for improvement). The tables and figures are also solid. The results do a fair job of matching the hypothesis. One area of improvement would be reframing aspects of the findings in terms of generally applicable recommendations. I discuss this more within the additional comments, which I hope is helpful!

Experimental design

The framework of the article is straightforward: exploring the applicability of a variety of common modeling approaches on a very impressive SCR dataset and comparing and contrasting estimates across approaches. The research gap that is addressed is about if non-Euclidean models (and finite-mixture models) should be used in instances where there is little to no expectation of landscape resistance from some aspect of the habitat. This is a fair question, and the authors explore the question with a solid dataset. Again, the SCR dataset is very impressive, and the authors clearly allocated a lot of computational effort to this project. The methods were sufficient most of the time. However, the authors should focus on double-checking how the details of their modeling approach align with standard recommendations from SCR modeling (related to spatial trap configuration, state-space modeling methods, and inclusion of a behavioral response), as it appeared there might be some inconsistencies, which I expanded upon within the additional comments.

Validity of the findings

There were doi references to the SCR dataset, and spatial covariates were referenced, as well. The study clearly uses a large dataset for an empirical investigation of a topic that has been studied with simulation studies as well as most often smaller empirical datasets, so it's interesting to see an application of ecological distance modeling here.

Additional comments

My most important (more major) comments are on lines:
1) L196
2) L195-196, L212, and L214
3) L300
4) L456


All comments:

Abstract

L17 - Here you are referencing multiple sources of heterogeneity, so consider rewording to: These types of heterogeneity can cause bias…”

L21 - Between these two sentences, it would be helpful to have a sentence that briefly describes the goal of this study.

L24 - This is an interesting result, but at this point in the Abstract we have no context as to why finite mixture models were used. That context isn’t provided until L28.

L24 - It’s unclear what “scale” is really referring to here. The resolution of the covariates? The size of the study area?

L30 - The real key to estimating landscape connectivity is in the number of spatial recaptures, not captures per individual, and it could be helpful to reframe that here. I expand on this in my later comments.

L31 - This sounds rather circular. Consider instead saying: “In our study areas, where we expected that landscape features do not seriously constrain movements,…”. Also, it’s unclear if “our study areas” refers to all 52 study areas or some subset of them.

L32 - From investigations of this issue, there is a recommendation of the maximum resolution of a landscape covariate, and it should be 0.5-1x the SCR spatial scale of detection parameter. It could be helpful to frame this sentence in reference to the standard recommendation. I expand on this in my later comments.

L36 - This sentence is rather long and becomes slightly confusing, especially what “it” is referring to in this last line. Consider rephrasing.


Introduction

L81 - Unclear what is meant by a dataset needing to have reliable application. Clarification would be very helpful.

L85 - This sentence is confusing, I think you’re trying to say it’s easier to model heterogeneity when you know the underlying cause of it, but the way the sentence is currently phrased doesn’t get that point across easily.

L112 - I think what you’re trying to say here is that the approach has significant potential, not that it’s potentially significant, is that correct?

L113 - Jumping into how ecological distance is calculated is jarring. Consider stepping back and fleshing out what is meant by ecological distance. From there, you could go into why it’s useful (which you discuss already, in that is more interpretable).

L135 - Typo: dropped the “s” on “corridor”

L137 - Typo: extra “s” on “model”

L132-L140 - After reading this paragraph several times, I’m not quite sure what point you’re trying to make. It seems like you might be using the jaguars as a counterexample for a hypothetical case of not knowing what structures movement. But the setup for that leads the reader to expect an example of the opposite. Consider revisiting this paragraph.


Methods

L173 - This seems like a diverse conifer community, with conifers of different canopy structures and resulting penetrability to large animals. Could this create any heterogeneity in movement?

L196 - This is a baited sampling design, which could lead to a behavioral response (i.e., “trap happiness”), but it appears the models used did not account for this. Can you expand on this choice in the text? See Augustine et al. (2014) for example.

L195-L196 - No mention here of trap design relative to standard recommendations. Is this trap spacing equivalent to the standard recommendation of ~2x the spatial scale of detection? SCR is very sensitive to adequate sampling design, so this must be addressed directly. Also, was it a grid of 40-45 traps (with all inter-trap distances equivalent), or a more heterogeneous design? A figure of some of the designs could be helpful.

L203 - 204 - Can you provide information on what effect this “discarding” might have on SCR estimates? Or should it be completely random such that all it does is lowers the detection intercept (g0)?

L212 - The resolution of the integration mesh should be equivalent to ~0.5-1x the spatial scale of detection, and the buffer should be ~3x the spatial scale of detection. As with trap spacing, you should reference standard recommendations here. This is all discussed in the Royle et al. book Spatial Capture-Recapture.

L214 - At this point, you have stated:
- Trap spacing of 1.5km, implying a sigma of 0.75km
- Integration grid of 1km resolution, implying a sigma of 1-2km
- State-space buffer of 15km, implying a sigma of 3km
These values appear to not match up with standard recommendations. If the traps are set according to standard recommendations, then the integration grid should be 0.375-0.75km in resolution, and the state-space buffer should be 2.25 km. This needs to be addressed — how did you choose the values for these three design components? What is the biological justification? I did not see estimates of sigma in the text for comparison, either.

L280 - To what is “the coarser scale described above” referring? Perhaps you mean “described below”, in the next section: Spatial resolution.

L292 - I could be misremembering, but I believe that alpha2 values of zero should be ok, and a value of zero just makes the scaling of Euclidean distance to ecological distance equal to 1, such that they’re equivalent. Further, if you don’t allow for zero-valued cost values, then how do you estimate negative cost values (e.g., a covariate class that relatively amplifies movement)? For example… cost(step1, step2) = (exp(alpha2*z(step1)) + exp(alpha2*z(step2)))/2 … cost = (exp(0*1) + exp(0*1))/2 … 1 + 1 / 2 = 1. So, in this way, the alpha2 parameter is converted to a cost, and (as you noted) the inverse of cost is connectivity.

L300 - It could be more generally applicable if you tested resolutions relative to sigma values, rather than arbitrary ones (e.g., 1-quarter sigma, 1-half sigma, 1-sigma, vs. 60, 120, 240, 520, 960).


Results

L328 - Please define “movements between traps” specifically.

L351-352 - Resolving the scale issue for the design relative to sigma should make it more clear what covariate resolutions should be useful to test, and it could be that the resolution you were testing at was far smaller than necessary (recommendation is 0.5-1x sigma), which would be bumping up your computation time substantially. Also, given that the other design components suggest a sigma of around 0.75-1 km, the 15km buffer is likely far more than necessary, which would also dramatically increase computation time.

L372 - It would be helpful to streamline terminology. Overall, you use: non-Euclidean distance, ecological distance, cost, and landscape resistance. Most of these are fitting in the way they are used, but pointing out their relative meaning would be very helpful for the reader.

Discussion

L388 - “…would account for *otherwise* unmodelled detection heterogeneity…” might be better.

L389 - It seems like this sentence should be emphasizing that you expect non-Euclidean models to detect heterogeneity even where it might not be obvious (no dramatic structure), is that correct? Consider revisiting this sentence to rephrase it.

L396 - Focusing on movement recaptures is preferred over detections per individual since the former is more informative about movement and the ability to estimate alpha2.

L404 - “Contrary to past applications” I’m not sure exactly the point that you’re making here. Are you arguing that the listed past applications should not have used non-Euclidean models? Or that non-Euclidean models should not be used outside of situations where you have a good expectation of an important spatial covariate (as the listed examples did have)?


Conclusions

L456 - I noted that the emphasis should be on movement/spatial recaptures in L396, and it seems you’ve recognized that here. Making that point and shifting emphasis in that way throughout the paper would be very helpful since movement recaptures are the most important data for estimating how movement is affected by landscape structure.


Tables & Figures

Table 2 - Is “movement model” referring to what spatial covariate was used to estimate cost? If so, consider saying “spatial covariate” instead. Also, it would be helpful to clarify what the values in the table represent exactly.

Figure 1 - Plotting out the locations is helpful, but it doesn’t appear that there is much broad-scale spatial structure to density estimates or RSE. Perhaps there might be a better way to represent this?


References:

Augustine, B. C., Tredick, C. A., & Bonner, S. J. (2014). Accounting for behavioural response to capture when estimating population size from hair snare studies with missing data. Methods in Ecology and Evolution, 5(11), 1154-1161.

---

## Round 0.2 · Minor Revisions

Dear Dr. Marrote and Dr. Northrup,

You are almost done. Please proceed to improve the manuscript according to what was raised by reviewer #3 and I believe you will have this manuscript accepted.

Sincerely,
Daniel Silva

Reviewer 1 ·

Basic reporting

Well written, good literature review, tables and figures relevant, and hypotheses addressed.

Experimental design

Experimental design sufficient and rigorous analysis of results. Methods well described.

Validity of the findings

Conclusions well stated and not outside the scope of their analysis and data.

Additional comments

The paper “Explaining detection heterogeneity with finite mixture and non-Euclidean movement in spatially explicit capture-recapture models” by Marrotte et al. is an analysis of finite mixture, non-mixture, and non-Euclidean distance (non-ED) models to address individual heterogeneity in detection rates of black bears in Ontario. I previously reviewed this manuscript and now comment on the revision.

L41. “Data” is plural (change to “data are available”).
L45. Really a nice introduction.
L84. I am not sure what the conventions are for this journal but this list of citations is not chronological nor alphabetical.
L204. By “scale of female movements” do you mean σ? Please define.
L516. I think it is important to point out that factors other than movement can cause individual capture heterogeneity that the non-Euclidean models do not assess. Bears commonly approach hair traps but do not go in them; I think that some animals innately avoid entering traps and the non-Euclidean models will not identify those individuals. However, the finite mixture models might. I realize that random effects models are outside the scope of your paper but individual random effect models show promise for accounting for individual heterogeneity in non-SCR capture-recapture models (White, G. C., and E. G. Cooch. 2017. Population abundance estimation with heterogeneous encounter probabilities using numerical integration. Journal of Wildlife Management 81P:322–336.). This has not been incorporated into maximum likelihood SCR models yet but that might make a good recommendation.
As an aside, I have found that finite mixture models are often affected by a few outlier observations. A very large σ can often be associated with some large movements from a few individuals and re-running the model without those outlier captures can reveal whether they are the source for one of the mixtures. I suspect that your very small estimates of σ might be due to multiple captures of individuals at only 1 trap, maybe because a trap is isolated from the others. You might try removing a few of those repeat captures at single traps to see how much the mixtures are affected. It could be that the mixture models are performing as they should and the problem is with study design.

In general, I am satisfied with how you addressed my concerns expressed in my earlier review.

Reviewer 3 ·

Basic reporting

The authors substantially revised the manuscript for clarity. All else has remained satisfactory.

Experimental design

The design itself has remained the same, though the justification is seriously improved, and the design is solid.

Validity of the findings

The presentation of the findings is now more precise, and this is a meaningful contribution.

Additional comments

I'm satisfied with the authors' responses to my comments. Below, I elaborate on this regarding particularly important comments. Line number references correspond to those in the rebuttal document. I'm not requiring the authors to respond to these four points; the comments are just meant to clarify.

- The miscommunication between myself and the authors (on many points) was not that I ignored differences between sexes, but that it was not clear what the sigma value was for either sex based on what was written in the manuscript. That information should have been conspicuous. I see that the authors have included that information now explicitly (<1km for females, <2km for males).

- The reference to the Royle et al. book was simply a citation of this recommendation in the literature (a standard practice). I was not implying it need be cited in their manuscript.

- My comment on line L212 included a typo, apologies: the state-space buffer of 15 km should imply a sigma of 5km, not 3km (simple reversal). But, my point remained valid as I had explained it initially (SS buffer >= 3 x sigma, typically).

- In general response to my L212 comments: Again, sigma values for the sexes were not included obviously in the main text, and back-calculating them from design choices suggested incongruity that would have been, at the very least, confusing for the reader. I'm satisfied with the response from the authors and the additional information they included in the manuscript.

---

## Round 0.3 · accepted · Accept

Dear authors!

Congratulations! Your manuscript was formally accepted for publication in PeerJ. Well done!